# Hand Carriage of Yeast in Student of Medicine, Nursing, and Medical Laboratory Science: Impact of Infection Control Measures

**DOI:** 10.3390/microorganisms12091907

**Published:** 2024-09-19

**Authors:** Víctor Silva, Ceidy Silva, Coral Silva, Rodrigo Gacitúa, Hernán Salas, Neftalí Guzmán, Claudio Alburquenque, Viviana Silva-Abello

**Affiliations:** 1VSV-Consulting-LATAM, Pucón 4920000, Chile; 2Unidad de Medicina, Hospital Santo Tomas de Limache, Valparaíso 2240421, Chile; ceidy.alejandra@gmail.com; 3Facultad Ciencias de la Vida, Universidad Andrés Bello, Santiago 8370251, Chile; coralsilvafebre@gmail.com; 4Escuela de Tecnología Médica, Facultad de Medicina, Universidad Mayor, Santiago 8580745, Chile; rgo.unlp.md@gmail.com (R.G.); claudio.alburquenque@umayor.cl (C.A.); viviana.silva@umayor.cl (V.S.-A.); 5Unidad de Medicina, Hospital de Purranque, Osorno 5290000, Chile; eldocsalas@gmail.com; 6Laboratorio de Investigación en Salud de Precisión, Departamento de Procesos Diagnóstico y Evaluación, Facultad de Ciencias de la Salud, Universidad Católica de Temuco, Temuco 4810399, Chile; nguzman@uct.cl

**Keywords:** hand yeast carriage, healthcare students, *C. parapsilosis*, control infection, antifungal susceptibility

## Abstract

We studied yeast hand carriage of 260 healthcare students. We isolated yeasts in 27 students (10.4%), without differences between medicine, nursing, and medical laboratory science programs and gender. A significant lower prevalence of carriage was shown in the clinical cycle (2.7%) compared to the basic cycle (13.5%) (*p* = 0.022) and the preclinical cycle (13.5%) (*p* = 0.014). Increased handwashing frequency and the use of alcohol gel and antiseptic soap decreased yeast carriage. Students who applied moisturizing hand cream two or more times a day had a lower frequency of yeast carriage (3.4%) than those who did not use it or used it once a day (16.5%), showing a significant difference (*p* = 0.016). The most prevalent species was *C. parapsilosis sensu stricto* (81.5%), followed by *Meyerozyma guilliermondii* (*C. guillermondii*) (7.4%), *Trichosporon mucoides* (7.4%), and *R. mucilagenosa* (3.7%). One case showed mixed carriage of *C. parapsilosis* and *C. albicans*. All strains were sensitive to voriconazole, caspofungin, and anidulafungin. This study shows hand carriage of yeast in health students, mainly by *C. parapsilosis*, and the frequency of infection control measures and moisturizing hand cream is associated with colonization control.

## 1. Introduction

Species of *Candida* spp. are important agents of healthcare-associated infections (HAIs), and skin yeast colonization often precedes invasive acquired infections, primarily causing catheter-associated mycosis, invasive candidiasis, and sepsis associated with central venous catheters or medical devices, especially in patients admitted to intensive care units, onco-hematology units, and neonatology [1,2,3,4,5].

Globally, HAIs caused by yeasts are mainly due to *C. albicans*, and the second most prevalent species varies by geographic area, type of candidiasis, and study period. *C. parapsilosis* has been reported as the second most frequent agent of candidemia in the Latin American and African region from the 2000s to present, while *C. glabrata* is prevalent as etiological agent candiduria [2,4,5,6,7,8].

Azoles and echinocandins are globally used to treat deep fungal infections. However, resistance to these antifungal drugs has increased in many of the *Candida* species [9]. In a recent meta-analysis, the lowest susceptibility in China was detected for the azole group; fluconazole susceptibilities were reported as follows: *C. parapsilosis* (93.25%), *C. albicans* (91.6%), *Nakaseomyces glabrata* (*C. glabrata*) (79.4%), *C. tropicalis* (77.95%), *Meyerozyma guilliermondii* (*C. guilliermondii*) (76%), *C. pelliculosa* (50%), and *C. auris* (0%). Amphotericin B and anidulafungin were the most susceptible drugs for all *Candida* species [10]. In addition, *Candida* species can form biofilms on implanted medical devices and at the host tissue, and these *Candida* biofilms are inherently resistant to traditional antifungal therapies and the immune system; therefore, biofilm-associated candidiasis are a huge clinical challenge [11].

*Candida auris* is a multidrug-resistant fungal pathogen that has been reported in more than 40 countries. Most importantly, *C. auris* is the first fungal pathogen to show pronounced and sometimes untreatable clinical drug resistance to all known antifungal classes, including azoles, amphotericin B, and echinocandins [5,12].

*Candida* species, mainly *C. parapsilosis* and *C. auris*, have the potential to cause outbreaks in healthcare settings, due to their efficient transmission via skin-to-skin contact [12,13]. Recently, the WHO reported 19 fungal priority pathogens to guide research, development, and public health actions, placing *C. albicans* and *C. auris* as fungi from the critical priority group and *C. parapsilosis* and *C. tropicalis* as members of the high priority group according to the prioritization process, which is focused on fungal pathogens that can cause invasive acute and subacute systemic fungal infections [5].

*Candida* species, mainly *C. albicans*, can be part of the human microbiota, colonizing more frequently the mucous membranes of the intestinal tract (50% to 70%), oral cavity (30% to 50%), and vaginal mucosa (5% to 30%) [8]. In contrast, the presence of *Candida* spp. on the hands of the general population is low, with reported frequencies of less than 7% [13,14]. However, yeast colonization on the hands of healthcare workers ranges from 20% to 90%, being an important vehicle for the transmission of these agents in healthcare settings [15,16,17,18,19,20].

Close contact between healthcare workers and patients, as well as with healthcare-related items in a clinical or hospital setting, would favor colonization of the hands of healthcare workers. This was partially evidenced by a previous study from our group, which reported a 16% prevalence of yeast colonization on the hands of medical students, with significant differences between the first two years of university (basic cycle, 7%) and the last two years of study (clinical cycle, 30%). The amount and diversity of species on the hands increased as students advanced in their training, associated with longer contact time with the hospital environment. In contrast, the frequency of colonization, amount of yeast, and diversity of species remained constant in the engineering student group [13]. These data were partially confirmed by Muango et al. (2017) in medical and nursing students, with a prevalence of hand yeast carriage of 11.2% and 9.2%, respectively [19].

Sixty years ago, the main clinically significant *Candida* species was *C. albicans*, and its colonization frequency on hands ranged from less than 1% in community individuals to 17% on the hands of healthcare workers [14]. Later studies reported a broader spectrum of yeast species on the hands of healthcare workers and students, predominantly *C. parapsilosis sensu stricto* [13,15,17,18,19,20,21].

Permanent and transient yeast colonization on hands is a potential vehicle for transmission to patients, both directly via skin-to-skin contact and indirectly through the contamination of medical devices, contributing to increased HAIs and outbreaks [4,8,15,17,18,20]. Handwashing can reduce the presence of yeasts by 50%, but if an antiseptic such as chlorhexidine, povidone–iodine, or alcohol gel is added, colonization decreases significantly, reaching 10.5%, 18.7%, and 21.1%, respectively [21].

Considering that the hands of healthcare workers play an important role as a vehicle for the exogenous transmission of yeasts to patients, and the constant reporting of HAIs caused by yeasts, it is essential to know the frequency of yeast carriage on the hands of individuals associated with hospital services. Therefore, we performed this study to determine the prevalence of yeast carriage on the hands of students in the fields of medicine, nursing, and medical laboratory science, in relation to variables such as study cycle, frequency of application of biosecurity, and infection control measures, as well as the prevalence of yeast species and antifungal susceptibility patterns, to update the information that supports future recommendations for the prevention and control of cross-transmission of these opportunistic agents.

## 2. Materials and Methods

We performed a descriptive, prospective, and analytical study to determine the frequency of hand yeast carriage in healthcare students, how infection control measures influence the colonization rate, the prevalence of yeast species, and the antifungal susceptibility profile of the isolated strains. In 2019, students from medicine, nursing, and medical laboratory science programs at a private university in Santiago de Chile were informed and invited to participate in the study through a consent letter.

Inclusion criteria: healthy, regular medical, nursing, or medical laboratory science students. Exclusion criteria: students with any type of skin lesion or those using antifungal medication.

A survey was administered to each volunteer, which included questions about age, gender, program, university study cycle (basic-B (1st and 2nd year), preclinical-PC (3rd and 4th year), or clinical-C (5th year)), the average daily frequency of clinical handwashing, alcohol gel use, and moisturizing cream use, among others.

For sample collection, each student was asked to perform a simple hand wash for 1 min using 10 mL of sterile saline solution kept in Falcon tubes. The saline solution was placed in the students’ palms, who then performed handwashing for 1 min. The sample was collected using a swab, which was vigorously rubbed on the palms of both hands. The swab was then placed in a conical tube with 0.5 mL of sterile saline for transportation to the laboratory, at 4 °C to 6 °C, according to Silva et al. 2003 [13]. The sample was immediately received in the laboratory, where the quality of the sample, transportation conditions, and documents such as the signed consent form, completed survey, and time of sample collection were verified. The samples were plated on two Petri dishes, one with Sabouraud dextrose agar plus chloramphenicol and the other with chromogenic agar (Chrom-agar *Candida*), within 4 h of collection. The plates were incubated at 36 ± 1 °C for 72 h.

Yeast identification was performed using standard methodology. Briefly, after observing and analyzing the colony characteristics, a germ tube test in fresh human plasma incubated at 36 ± 1 °C for 2.5 h was performed for suspected *Candida* spp. strains, and microcultures in rice agar with 1% tween 80 incubated at 30 °C for 72 h were performed for suspected *Candida* spp. and *Trichosporon* spp. [8]. Biochemical tests were performed on all yeast strains using API 32C galleries (Biomerieux) incubated at 30 °C for 24 or 48 h, with the Api-Web^TM^ software version 4.0 used for interpreting the bio-code.

Identification of species in the *C. parapsilosis* complex (*C. parapsilosis sensu stricto*, *C. methapsilosis*, and *C. orthopsilosis*) was performed by genotyping using the secondary alcohol dehydrogenase (SADH) gene polymerase chain reaction (PCR) amplification strategy, which generates a 716 bp fragment in the *C. parapsilosis* complex species. This fragment was digested with the BanI restriction enzyme to molecularly discriminate between *C. parapsilosis sensu stricto*, which generates two bands of 521 bp and 196 bp; *C. orthopsilosis*, which presents a 716 bp band; and *C. methapsilosis*, which generates four bands of 370 bp, 188 bp, 93 bp, and 60 bp [22,23]. Briefly, DNA was extracted with the Wizard Genomic DNA kit (Promega, Madison, WI, USA) from the culture of the strain identified as *C. parapsilosis* by phenotypic methods. Primers S1F (5′-GTTGATGCTGTTGGATTGT-3′) and SIR (5′-CAATGCCAAATCTCCCAA-3′) were used to amplify the SADH gene in a T100 thermal cycler (Bio-Rad, Hercules, CA, USA). Amplified products and a 100 bp DNA ladder (Invitrogen Life Technologies, Carlsbad, CA, USA) were electrophoresed on 1.5% agarose gels, visualized under ultraviolet light, and analyzed on a photo documenter (Bio-Rad, Hercules, CA, USA). The 716 bp PCR fragments were purified with a specialized kit (Qiagen, Germantown, MD, USA), and digestion of the purified PCR product was carried out for 120 min at 37 °C using the restriction enzyme BanI (Thermo Scientific, Waltham, MA, USA). Digestion products were subjected to electrophoresis on 2% agarose gels (Invitrogen Life Technologies, Carlsbad, CA, USA).

Confirmation of *C. albicans* or *C. dubliniensis* was performed by PCR with specific primers designed previously by our group from 18S, ITS1, and ITS2 of the main six clinically relevant *Candida* species, as indicated in Table 1 [8]. In summary, DNA was extracted with the Wizard Genomic DNA kit (Promega, Madison, WI, USA) from the culture of the strain identified as *C*. *albicans* by phenotypic methods. The PCR reaction included 25 pmol of primers and 3 μL of yeast DNA in a total volume of 25 μL. The PCR conditions were one cycle at 95 °C for 3 min, followed by 40 cycles at 94 °C for 1 min, 60 °C for 2 min, and 72 °C for 3 min, with a final cycle at 72 °C for 7 min. Amplification was performed in a T100 thermal cycler (Bio-Rad, Hercules, CA, USA), and the amplified products were loaded onto 2% agarose gels containing ethidium bromide (0.5 mg/mL). A 100 bp DNA ladder was used as a molecular size (Invitrogen Life Technologies, Carlsbad, CA, USA).

The test of antifungal susceptibility to voriconazole, anidulafungin, and caspofungin was performed using the agar diffusion method following CLSI M44-A2 [24] recommendations. Briefly, from each yeast strain isolated on Sabouraud dextrose agar at 36 ± 1 °C for 24 h, a 0.5 McFarland inoculum was prepared using a turbidimeter to seed 1–5 × 10^5^ CFU/mL on Müller–Hinton agar plus 2% glucose and 0.5 µg/mL Methylene Blue for gradient strip testing (E-Test). Plates were incubated at 36 ± 1 °C for 20 to 24 h, and visual reading of the inhibition ellipse was performed. In vitro interpretative criteria were analyzed according to CLSI M27M44S [25].

The reference strains used for quality control were *C. albicans* ATCC 90028, *C. dubliniensis, Pichia kudriavzeveii* (*C. krusei*) 6258, and *C. parapsilosis* ATCC 22019.

Statistical analysis was performed using Fisher’s test, relative risk, and the odds ratio, with a 95% confidence level or power for significance.

## 3. Results

From a total of 300 students invited to participate, we studied hand yeast carriage in 260 students (167 or 64.2% female) from medicine (78), nursing (89), and medical laboratory science (93) programs.

Table 2 shows that the prevalence of hand yeast carriage was detected in 27 students (10.4%), with no significant differences among the three healthcare programs, although nursing students showed a slightly higher carriage (13.5%), while medical laboratory science students showed the lowest carriage (7.5%). The distribution of hand yeast carriage by gender was slightly higher in women with 20 carriers (12%) compared with 7 men carriers (7.5%), with no statistical difference. In summary, yeast colonization was observed on the hands of students from all study programs and in both genders, showing a slightly higher prevalence of colonization in the hands of nursing students.

Table 3 shows a significant difference in yeast colonization on students’ hands according to the university study cycle. Yeasts carriage was less prevalent in the clinical cycle with 2 carriers (2.7%), compared to 12 carriers (13.5%) in the basic cycle (*p* = 0.022) and 13 carriers (13.5%) in the preclinical cycle (*p* = 0.014). Notably, no yeast was detected on the hands of students in the clinical cycle of the medicine and medical laboratory science programs. These data are highly relevant, as they indicates that, as students in medicine, nursing, and medical laboratory science progress through their respective curricula towards the clinical cycle, the prevalence of yeast colonization on their hands decreases significantly. This is most likely due to the implementation of improved hygiene practices, which will be further discussed in the subsequent results.

Table 4 shows a tendency towards a decrease in the frequency of yeast carriage on hands according to handwashing frequency, decreasing from 15.8% in students who wash their hands three or fewer times a day to 7% in students who wash their hands seven or more times a day, with a relative risk of 2.13 and an odds ratio of 2.5. This tendency is similar to the use of antiseptic soap, as a 6.8% yeast carriage rate was observed in individuals who use this product compared to 14.2% in those who do not use it. Although no statistical difference was observed, likely due to the sample size, it is important to note that a higher frequency of daily handwashing and the use of antiseptic soap were associated with reduced yeast colonization on students’ hands (*p* > 0.05).

Table 5 shows that there was a strong tendency towards a decrease in yeast carriage frequency among students who use hand sanitizer (alcohol gel) five or more times a day (3.2%), compared to students who do not use hand sanitizer (14.4%), with a relative risk of 2.2 and an odds ratio of 5.1, but without a statistical difference (*p* > 0.05). This result demonstrates an inverse relationship between alcohol gel use and the frequency of yeast colonization on students’ hands.

Table 6 shows a significant decrease in yeast carriage among students who apply hand moisturizing cream with a frequency of four or more times a day (0%) or two to three times a day (3.4%), compared to those who do not use it or apply it only once a day (16.5%), with a relative risk of 2.6 and an odds ratio of 5.5 and *p* = 0.016. This information is highly relevant, as it statistically confirms that the frequent use of moisturizing cream on the hands of medical, nursing, and medical laboratory science students is associated with a lower prevalence of yeast colonization. These results indicate that the combination of frequent hand washing with antiseptic soap, plus alcohol gel and moisturizing cream, are appropriate measures for the control of yeast colonization on the hands.

The yeast species most predominantly on the hands of nursing, medicine, and medical laboratory science students was *C. parapsilosis*, recovered in 22 out of 27 students (81.5%), followed by *Meyerozyma guilliermondii* (*C. guilliermondii*) in 2 students (1 from nursing and 1 from medicine), *Trichosporon mucoides* in 2 medical laboratory science students, and *R. mucilagenosa* in 1 nursing student. One medical laboratory science student had simultaneous carriage of *C. parapsilosis* and *C. albicans* on their hands (Table 7).

The 22 strains identified as part of the *C. parapsilosis* complex, using the mycology conventional method, were amplified by PCR, obtaining a 716 bp fragment, which is characteristic of the secondary alcohol dehydrogenase gene—SADH. Figure 1 shows the amplified fragment of this gene in 18 strains of this yeast complex obtained from the hands of nursing, medicine, and medical laboratory science students.

According to the Ban I restriction pattern of the amplified SADH gene fragment (716 bp), all isolates of the *C. parapsilosis* complex strains were molecularly discriminated as *C. parapsilosis sensu stricto*, as evidenced by the two bands of 521 bp and 196 bp that characterize this species (Figure 2). No profiles associated with *C. methapsilosis* or *C. orthopsilosis* were detected.

In this study, one medical laboratory science student presented yeast hand colonization by two species of *Candida*, identified as *C. parapsilosis* and *C. albicans* by classical identification methods in mycology. Due to the phenotypic similarity between both species, a specific molecular method was applied to confirm the identification of the *Candida* species. The species-specific PCR confirmed the identification of *C. albicans* by generating a 724 bp fragment (*C. dubliniensis* generates a 354 bp amplicon) (Figure 3).

As shown in Table 8, all isolated yeasts were sensitive to voriconazole, caspofungin, and anidulafungin, with MIC90 values of 1 µg/mL, 2 µg/mL, and 1 µg/mL, respectively. These data are important since they show that all isolated yeasts, both *Candida* and *Trichosporon* species, are sensitive in vitro to these three antifungals.

## 4. Discussion

*Candida* spp. is an opportunistic pathogen that causes superficial and invasive infections with nosocomial outbreaks without strict hygiene protocols. During hospitalization, colonization by *Candida* spp. is frequent and often precedes invasive hospital-acquired infections. Recently, it was reported that in neonates, skin colonization by *Candida* spp. was 45% [4], as well as in adult patients (9825), where the prevalence of *Candida* colonization reached 40% (3886), and 4.7% (462) developed invasive candidiasis. Meta-analysis indicated that critically ill patients with sepsis who are colonized with *Candida* are more likely to develop invasive candidiasis (odds ratio 3.32; 95% CI 1.68–6.58) compared with non-colonized patients [3].

Our study shows that the prevalence of yeast carriage on the hands of nursing, medicine, and medical laboratory science students was 10.4%, being slightly higher in nursing students at 13.5% and lower in medical laboratory science students at 7.5%. The distribution of yeast carriage by sex was slightly higher in women (12%) than in men (7.5%), with no statistical difference. These data are similar to those previously published for medicine students at 16% [12] and for nursing and medicine students at 9.9% and 11.2%, respectively [19]. The presence of yeasts on the hands of healthcare students is higher than reported for the general community [13,14,15], but lower than detected among hospital personnel, with carriage rates ranging from 20% to 90% [6,15,16,17,18,19,20,21]. Dalben et al. [15] showed that 89.47% of healthcare professionals evaluated were colonized by *Candida* species on their hands. On the other hand, Yildirim et al. [18] reported that 34.1% of the people analyzed carried *Candida* spp. on their hands: 30.7% were nurses, 25.8% were resident doctors, 28.6% were laboratory workers, 84.6% were dining room personnel, and 43.3% were officers.

In this study, the hand yeast carriage in healthcare students was significantly less prevalent among clinical cycle students (2.7%) compared to younger students in the basic (13.5%) and preclinical (13.5%) cycles. These results differ from previous reports, which indicated that the prevalence of yeast carriage in medical and nursing students increases progressively as they advance to the clinical cycle and spend more hours in the hospital environment [13,19]. Our results showing lower yeast carriage prevalence in the clinical cycle group may be explained by the greater adherence to clinical and infection control measures among advanced students, such as hand washing with antiseptic soap, plus alcohol gel and moisturizing cream use. We found that the frequency of yeast carriage decreased from 15.8% in students who wash their hands three or fewer times a day to 7% in students who wash their hands seven or more times a day. This tendency in controlling yeast carriage is reinforced with the use of antiseptic soap, as yeast carriage in students using this product is 6.8% compared to 14.2% in those who do not use it, along with frequent use of hand sanitizer or alcohol gel (3.2%) compared to students who do not use hand sanitizer (14.4%). Our results confirm the findings previously reported by Yildirim et al., who indicated that handwashing can reduce yeast presence by 50%, while the use of antiseptics such as chlorhexidine, iodine povacrylate, or hand sanitizer significantly decreases yeast carriage on hands to frequencies of 10.5%, 18.7%, and 21.1%, respectively [21], as well as chlorhexidine soap, which reduces microbial load when cleaning the skin [26].

We also detected that frequent use of moisturizing hand cream significantly decreases yeast carriage in students who apply this product two to three times (3.4%) and four or more times a day (0%) compared to those who do not use it or apply it only once a day (16.5%). This result may be partly explained by the application of moisturizing cream on the hands being associated with more frequent hand hygiene and the protective role of the moisturizing cream on keratinocytes of the skin surface, thus contributing to the integrity of the skin’s mechanical barrier, which can be affected by the use of latex or vinyl gloves and chemical antiseptics [17,27].

The predominant isolated species from the hands of nursing, medicine, and medical laboratory science students was *C. parapsilosis sensu stricto* (81.5%), confirmed molecularly by the restriction pattern with the Ban I enzyme of the amplified SADH gene fragment, affirming the usefulness of this genotyping technique by PCR [22,23]. Additionally, *Meyerozyma guilliermondii* (*C. guilliermondii*), *Trichosporon mucoides*, and *R. mucilagenosa* were isolated less frequently. The predominance of *C. parapsilosis* on the hands of healthcare students observed in this study was higher than reported in previous studies on hand yeast colonization among healthcare-related individuals, which ranged from 9.88% to 45% [6,15,17,18,19,20].

In many countries, especially in Latin America, *C. parapsilosis* has been reported as the second most frequently isolated fungal agent causing candidemia in both pediatric and adult patients over the past two decades [2,7,8], indicating the high level of circulation in hospital environments. In the early 2000s, a surveillance program for invasive fungal infections was implemented in Chile. The program reported that nosocomial fungal infections accounted for 83.5% of bloodstream infections (BSI), with *C. albicans* (48.1%), *C. parapsilosis* (17.7%), and *C. tropicalis* (13.9%) being the most frequently isolated pathogens from blood cultures [2]. However, between 2006 and 2008, a significant epidemiological change in yeast prevalence of candidemia was observed in Chile. *C. albicans* remained the most prevalent species (55%), while the prevalence of *C. parapsilosis* decreased to 15%. In contrast, *C. glabrata* increased to 10%, equaling the frequency of *C. tropicalis* (11%), compared to data from the early 2000s [8]. Nucci et al. [7] developed the first large epidemiologic study of candidemia in Latin America, with an incidence of 1.18 cases per 1000 admissions and varying across countries, with the highest incidence in Colombia (1.96) and the lowest in Chile (0.33). *Candida albicans* (37.6%), *C. parapsilosis* (26.5%), and *C. tropicalis* (17.6%) were the leading agents.

Just one student presented simultaneous carriage of *C. parapsilosis* and *C. albicans*, confirmed by specific PCR, because *C. albicans* and *C. dubliniensis* are phenotypically similar [28]. The low frequency of *C. albicans* on the hands of community and healthcare personnel has been previously documented [13,14,15,19,20,21].

All yeasts isolated from students’ hands showed MIC values indicating sensitivity to voriconazole, caspofungin, and anidulafungin, consistent with other studies that have reported high sensitivity in *Candida* strains isolated from invasive infections [29,30]. However, this differs from the gradual increase in resistance to certain antifungals reported by some researchers [9,10]. It is important to note that in vitro sensitivity tests are performed with vegetative yeasts, whereas their condition can change radically when colonizing and forming biofilms on implanted medical devices such as catheters. These *Candida* biofilms are inherently resistant to traditional antifungal therapies and the host immune system [11].

The hands of healthcare personnel are a potential vehicle for transmitting this agent, highlighting the importance of understanding its carriage among healthcare-related individuals and implementing appropriate control measures to prevent infections caused by species of *Candida*, mainly *C. parapsilosis*.

Among the limitations of our study are the absence of a control group of students, the lack of quantitative culture analysis, and the exclusion of other antifungal agents, such as fluconazole and amphotericin B. Further studies are required to validate our findings, including the implementation of multivariate statistical analyses to assess infection control measures. Additionally, it is recommended to examine the long-term impact of the frequency of their application over time.

## 5. Conclusions

This study highlights yeast carriage on the hands of nursing, medicine, and medical laboratory science students, with *C. parapsilosis sensu stricto* being the most prevalent species and no resistance detected among the analyzed strains. The frequency of yeast colonization on hands significantly decreased in clinical cycle students, since they show high adherence and greater frequency of application of infection control measures such as handwashing, antiseptic soap, alcohol gel, and moisturizing hand cream.

## Figures and Tables

**Figure 1 microorganisms-12-01907-f001:**
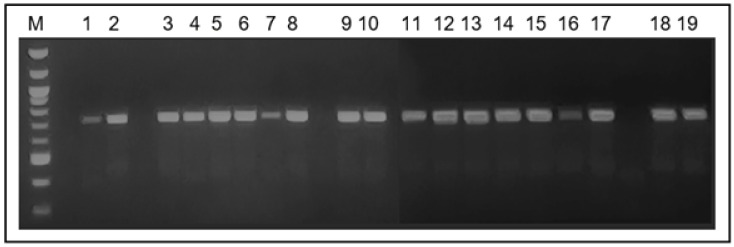
Amplification result of the secondary alcohol dehydrogenase (SADH) gene of the *C. parapsilosis* complex in 18 strains isolated from the student’s hands. M: 100 bp ladder. Lanes 1 to 18: *C. parapsilosis* strains isolated from the hands of students; lane 19: *C. parapsilosis* ATCC 22019 strains. All strains amplified the 716 bp fragment of the SADH gene.

**Figure 2 microorganisms-12-01907-f002:**
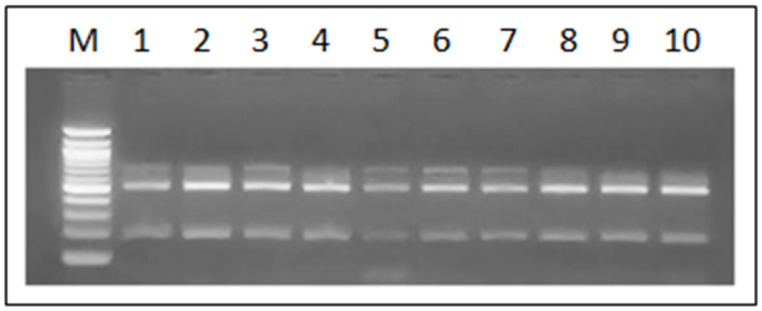
Results of Ban I digestion of the previously PCR-amplified SADH gene fragment (716 bp) in 9 strains of the *C. parapsilosis* complex isolated from the hands of health students. M: 100 bp ladder. Lane 1 *C. parapsilosis* ATCC 22019; lanes 2 al 10 *C. parapsilosis* strains isolated from the hands of students. All strains presented 2 bands of 521 bp and 196 bp.

**Figure 3 microorganisms-12-01907-f003:**
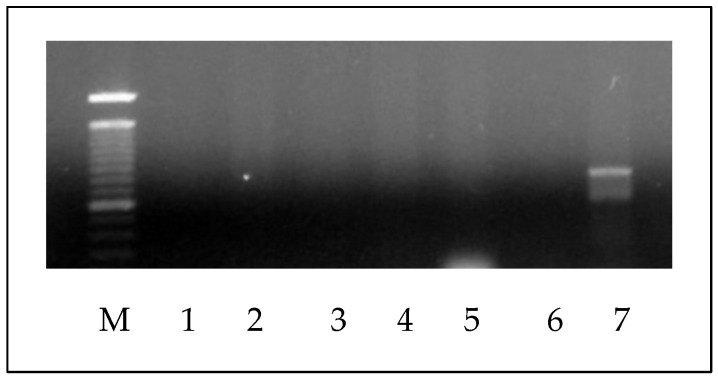
Result of *Candida* species-specific PCR showing a fragment of 724 bp of *C. albicans* strain. M: 100 bp ladder. Lanes 1 to 6 *C. parapsilosis*; lane 7 *C. albicans* with amplicon of 724 bp.

**Table 1 microorganisms-12-01907-t001:** Specific primer sequence used to identify the main 6 *Candida* species [8].

*Candida* Species	Primer Name	Specific Sequence 5′-3′	Position Gen	Fragment Size
*C. albicans*	Ca-F1 Ca-3RF	TACACTGACGGAGCGAGC AAAGCGATCCCGCCTTACC	18S—1455 bp ITS2—93 bp	724 bp
*C. dubliniensis*	Cd-2F Cd-3RC	TGTTCTGGACAAACTTGC CAAGCAATCTCCGCCTTAT	ITS1—31 bp IST2—97 bp	354 bp
*Nakaseomyces glabrata* (*C. glabrata*)	Cg-1F Cg-3RC	CCAAGTCCTTGTGGCTTG ACTGATATGGCCTACAAT	18S—679 bp ITS2—45 bp	1725 bp
*Pichia kudriavzeveii* (*C. krusei*)	Ck-1F Ck-2RC	TACCTATGGTAAGCACTG TTCGTTCCGCTCACGCAG	18S—634 bp ITS1—19 bp	1200 bp
*C. parapsilosis* complex	Cpc-1K Cpc-2RC	TCCGGATTGGTTTAGAGA GCCCCATATAGAAGGCCT	18S—1660 bp ITS1—72 bp	210 bp
*C. tropicalis*	Ct-2F Ct-3RC	GGTGGCGGGAGCAATCCT TGTGGTGGCCACTAGCAA	ITS1—20 bp ITS2—90 bp	340 bp

**Table 2 microorganisms-12-01907-t002:** Distribution of students with yeast carriage on hands by university program.

Yeast Cultures	Nursing	Medicine	Medical Lab Science	Total
Positive	12 (13.5%)	8 (10.3%)	7 (7.5%)	27 (10.4%)
Negative	77 (86.5%)	70 (89.3%)	86 (92.5%)	233 (89.6%)
Total	89 (100%)	78 (100%)	93 (100%)	260 (100%)

*p* > 0.05.

**Table 3 microorganisms-12-01907-t003:** Distribution of students according to hand yeast carriage and study cycle as basic, preclinic, and clinic by each university program.

Yeast Cultures	Nursing (N°) B-PC-C	Medicine (N°) B-PC-C	MTS (N°) B-PC-C	Total N°(%) B-PC-C
Positive	6-4-2	4-4-0	2-5-0	12 (13.5) *-13 (13.5) *-2 (2.7) *
Negative	24-23-30	26-26-18	27-34-25	77 (86.5)-83 (86.5)-73 (97.3)
Total	30-27-32	30-30-18	29-39-25	89 (100)-96 (100)-75 (100)

MTS: medical laboratory science. B: basic cycle (1st and 2nd year). PC: preclinical cycle (3rd and 4th year). C: clinical cycle (5th year). * C vs. B *p* = 0.022 and C vs. PC *p* = 0.014.

**Table 4 microorganisms-12-01907-t004:** Distribution of students according hand yeast carriage and daily handwashing frequency and use of antiseptic soap.

Yeast Cultures	Daily Handwashing Frequency				Antiseptic Soap	
	≤3	4–6	≥7	Total	No	Yes
Positive	3 (15.8%)	15 (13,5%)	9 (7%)	27 (10.4%)	18 (14.2%)	9 (6.8%)
Negative	16 (86.5%)	97 (86.5%)	120 (93%)	233 (89.6%)	109 (85.8%)	124 (93.2%)
Total	19 (100%)	112 (100%)	129 (100%)	260 (100%)	127 (100%)	133 (100%)

*p* > 0.05, relative risk IC 95% = 2.13 and odds ratio IC 95% = 2.5.

**Table 5 microorganisms-12-01907-t005:** Distribution of students according to hand yeast carriage and use of daily hand sanitizer (alcohol gel) frequency.

Yeast Cultures	No	1–2	3–4	≥5	Total
Positive	17 (14.4%)	7 (9%)	2 (6.1%)	1 (3.2%)	27 (10.4%)
Negative	101 (85.6%)	71 (91%)	31 (93.9%)	30 (96.8%)	233 (89.6%)
Total	118 (100%)	78 (100%)	33 (100%)	31 (100%)	260 (100%)

*p* > 0.05, relative risk IC 95% 2.2 and odds ratio IC 95%: 5.1.

**Table 6 microorganisms-12-01907-t006:** Distribution of students according to hand yeast carriage and frequency of daily use of moisturizing cream.

Yeast Cultures	No.	1	2–3	≥4	Total
Positive	11 (10.5%) *	14 (16.5%) *	2 (3.4%) *	0 (0%) *	27 (10.4%)
Negative	94 (89.5%)	71 (83.5%)	56 (96.6%)	12 (100%)	233 (89.6%)
Total	105 (100%)	85 (100%)	58 (100%)	100 (100%)	260 (100%)

Use of moisturizing cream 2 or more times a day vs. one time per day or less. * *p* = 0.016, relative risk IC 95% 2.6 and odds ratio IC 95%: 5.5.

**Table 7 microorganisms-12-01907-t007:** Distribution of yeast species isolated from the hands of nursing, medicine, and medical laboratory science students.

Yeast Species	Nursing	Medicine	MLS	Total
*C. parapsilosis senso stricto*	10	7	5 *	22 (81.5%)
*Meyerozyma guilliermondii*	1	1	0	2 (7.4%)
*Trichosporon mucoides*	0	0	2	2 (7.4%)
*Rhodotorula mucilagenosa*	1	0	0	1 (3.7%)
Total	12	8	7	27 (100%)

* One student with simultaneous carriage of *C. parapsilosis* and *C. albicans*.

**Table 8 microorganisms-12-01907-t008:** Susceptibility test from 28 yeast strains isolated from students’ hands.

	Voriconazole		Anidulafungin		Caspofungin	
	MIC μg/mL	S %	MIC μg/mL	S %	MIC μg/mL	S %
Range	0.5–1	100%	0.5–2	100%	0.125–1	100%
MIC 50	1	100%	1	100%	1	100%
MIC 90	1	100%	2	100%	1	100%

## Data Availability

Our research data are available at: https://drive.google.com/drive/folders/1HiKIvA5hNvu_x9u0_MOZSL0z9ry1rhlZ (accessed on 18 September 2024).

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
