# Peer review of "Hand Carriage of Yeast in Student of Medicine, Nursing, and Medical Laboratory Science: Impact of Infection Control Measures"

_microorganisms, 2024, doi:10.3390/microorganisms12091907_

Round 1

Reviewer 1 Report

Comments and Suggestions for Authors

Review of the manuscript microorganisms-3187150

Medical personnel's hand colonization by fungi as well as by bacteria is a very well-known problem, as the hands are recognized as the primary vehicle to facilitate the spread of pathogenic microorganisms in hospital settings and among inpatients.

The knowledge about colonization among students is important as it helps to provide novel and more adequate methods for hais prevention.

The manuscript seems to be interesting, but before it can be considered to be accepted for publication several minor and major changes must be applied.

In the introduction part, the cited literature describing the prevalence of hand colonization is dated from 1993 to 2015. I believe you can provide also more recent data.

All the Latin names must be italicized. Make sure that it is corrected throughout the whole paper.

In the alter part of the introduction you focus on the characteristics of some fungi, however, I would recommend focusing on the ones you found in your study and on the ones that are known to cause HAIs in your hospital or region.

Additionally, in this part, the other references cited are quite old, eg. References 13, 14 15, 17, and 20 are dated between 1998 and 2012. Of course, those references can be used, however newer ones should also be added. In case they are not available, clearly state it, and discuss.

In the materials and methods section, the sample collection procedure is not clear. Was it 1 minute or 20 seconds procedure? Please revise the line from 117 to 127.

Moreover please state if there were any inclusion or exclusion criteria for the study participants.

For the PCR procedures, starting from line 136, please provide all the details of the PCR reaction, equipment used, regents, etc.

Table 2, Please add the reference for primers used in the study, or describe its design.

Please explain why only 3 antifungals were used in the study.

The results section should be more elaborated, currently, it's just a description of what is presented in several tables and figures.

The discussion section is generally good, however, I would expect that this part is also more elaborated and the findings are confronted and discussed with data regarding the prevalence of HAIS caused by specific fungal species in the region of study. If such data are not available as a published report/scientific report, maybe have access to unpublished data? This would greatly improve the significance of your study results and would be beneficial for the Hais prevention program's implementation.

Kind regards,

/-/

Author Response

Response to Reviewer of the Manuscript MICROORGANISMS – 3187150

I agree with the reviewer's comments regarding the manuscript. The topic of microorganism colonization on hands is highly relevant, and understanding this issue among healthcare students is crucial for developing new and more effective prevention strategies and methods.

Response to Reviewer Comments

  1. In the introduction part, the cited literature describing the prevalence of hand colonization is dated from 1993 to 2015. I believe you can provide also more recent data.

R: We agree with the reviewer. A PubMed search was conducted using the terms 'yeast OR Candida AND carriage OR colonization,' yielding 5,510 papers from 1968 to 2024. We included three additional papers from 2021 and 2024 that link cutaneous Candida colonization with invasive mycoses in ICU patients and neonates. However, the majority of recent studies focus on oral and intestinal yeast colonization, which are not relevant to this research. The studies of interest regarding yeast or Candida spp. colonization or carriage on hands are cited within this manuscript."

  • Alenazy H, Alghamdi A, Pinto R, Daneman N. Candida colonization as a predictor of invasive candidiasis in non-neutropenic ICU patients with sepsis: A systematic review and meta-analysis. Int J Infect Dis. 2021 Jan;102:357-362. doi: 10.1016/j.ijid.2020.10.092. Epub 2020 Nov 3. PMID: 33157294.
  • Mabena FC, Olwagen CP, Phosa M, Ngwenya IK, Van der Merwe L, Khan A, Mwamba TM, Mpembe R, Magobo RE, Govender NP, Velaphi SC, Madhi SA. Bacterial and Candida Colonization of Neonates in a Regional Hospital in South Africa. Pediatr Infect Dis J. 2024 Mar 1;43(3):263-270. doi: 10.1097/INF.0000000000004177.
  • Dalben YR, Pimentel J, Maifrede SB, Carvalho JA, Bessa-Neto FO, Gomes JFS, Leite GR, Rodrigues AM, Cayô R, Grão-Velloso TR, Gonçalves SS. Early Candida colonisation impact on patients and healthcare professionals in an intensive care unit. Mycoses. 2024 Aug;67(8):e13786. doi: 10.1111/myc.13786
  1. All the Latin names must be italicized. Make sure that it is corrected throughout the whole paper.

R: We agree with the reviewer. All Latin names were placed in italics throughout the text.

  1. In the alter part of the introduction you focus on the characteristics of some fungi, however, I would recommend focusing on the ones you found in your study and on the ones that are known to cause HAIs in your hospital or region.

R: We agree with reviewer and change the focus of the information on C. parapsilosis.

  1. Additionally, in this part, the other references cited are quite old, eg. References 13, 14 15, 17, and 20 are dated between 1998 and 2012. Of course, those references can be used, however newer ones should also be added. In case they are not available, clearly state it, and discuss.

R: We agree with the reviewer. See response to point 1. We eliminate 3 older references (Clayton et al., 1966, Horn et al., 1988 and Sanchez et ala. 1993) and include a new one:

  • Dalben YR, Pimentel J, Maifrede SB, Carvalho JA, Bessa-Neto FO, Gomes JFS, Leite GR, Rodrigues AM, Cayô R, Grão-Velloso TR, Gonçalves SS. Early Candida colonization impact on patients and healthcare professionals in an intensive care unit. Mycoses. 2024 Aug;67(8):e13786. doi: 10.1111/myc.13786.
  1. In the materials and methods section, the sample collection procedure is not clear. Was it 1 minute or 20 seconds procedure? Please revise the line from 117 to 127.

R: The sample collection procedure was revised; ….a simple hand wash for 1 minute using 10 mL of SS…..

  1. Moreover please state if there were any inclusion or exclusion criteria for the study participants.

R: We agree with the reviewer and include inclusion and exclusion criteria. (line 112 to 114)

  1. For the PCR procedures, starting from line 136, please provide all the details of the PCR reaction, equipment used, regents, etc.

R: We agree with the reviewer and complement with more details this section of methods (from line 138).

  1. Table 1, Please add the reference for primers used in the study, or describe its design.

R: We include the reference in the title of table 1.

  1. Please explain why only 3 antifungals were used in the study.

R: We use the antifungals that we have available in the laboratory and that are used in systemic treatments of candidemia or invasive candidasis.

  1. The results section should be more elaborated, currently, it's just a description of what is presented in several tables and figures.

R: We agree with the reviewer and we work better on the results in all of his tables and figures, thus complementing the information provided in the results section.  

  1. The discussion section is generally good, however, I would expect that this part is also more elaborated and the findings are confronted and discussed with data regarding the prevalence of HAIS caused by specific fungal species in the region of study. If such data are not available as a published report/scientific report, maybe have access to unpublished data? This would greatly improve the significance of your study results and would be beneficial for the Hais prevention program's implementation.

R: We agreed with the reviewer and the discussion was carried out, complementing and comparing the results with data from the main references.

Reviewer 2 Report

Comments and Suggestions for Authors

Quality of English needs improvements. This includes grammar and spelling across several parts.

Please correct acknowledgements.

Line 17 please correct to “carriers”.

The structure and flow of information in the introduction is somewhat confusing, as it seems that it goes back and forth. Please consider restructuring to maintain a smooth flow.

Methods:

please mention timeframe of study.

From what I understand, there was also a survey among participants evaluating their practices. Please mention this and describe the content of the survey.

Results:

Please mention total number of students asked to participate, in order to yield a response rate.

Please mention the study level of the participants and describe what study year the 3 levels correspond to (B, PC, C).

Tables 4&5: what do the p-values refer to?

Tables 3&6: the analysis here is confusing; have the authors performed a bivariate comparison of all factors? The p-values seem to refer only to one comparison. Please describe and adjust the presentation of analyses.

Discussion

Authors mention biosafety and infection control practices among students. Please describe these measures.

Lines 312-318: could more frequent application of moisturizing cream also be indirectly associated with more frequent hand hygiene? Please comment.

Please add a paragraph acknowledging the limitations of the study.

Comments on the Quality of English Language

Quality of English needs improvements. This includes grammar and spelling across several parts.

Author Response

1.     Quality of English needs improvements. This includes grammar and spelling across several parts.

R: The manuscript was revised and improvements were made, both in the correction of some words and the grammar of several paragraphs.

2.     Please correct acknowledgements.

R: Was corrected

3.     Line 17 please correct to “carriers”.

R: Was change by “programs”

4.     The structure and flow of information in the introduction is somewhat confusing, as it seems that it goes back and forth. Please consider restructuring to maintain a smooth flow.

R: The introduction section was revised and improved

5.     Methods: please mention timeframe of study.

R: Timeframe of study was include (line 110)

6.     From what I understand, there was also a survey among participants evaluating their practices. Please mention this and describe the content of the survey.

7.     R: The content of the survey was described in methods (lines 116 – 119). “A survey was administered to each volunteer, which included questions about age, gender, the field of study, the university study cycle: basic (1st and 2nd year), preclinical (3rd and 4th year), or clinical (5th year), the average daily frequency of clinical handwashing, alcohol gel use, moisturizing cream use, among others”.

8.     Results: Please mention total number of students asked to participate, in order to yield a response rate.

R: the total number of student invited to participate was indicate (line 188)

9.     Please mention the study level of the participants and describe what study year the 3 levels correspond to (B, PC, C).

R: the study level was better described in method (lines 117 and 118) and in table 3 (line 215)

10.  Tables 4&5: what do the p-values refer to?

R: The p value refers to the analysis between the frequency of hand washing, use or not of antiseptic soap and frequency of use of alcohol gel.

11.  Tables 3&6: the analysis here is confusing; have the authors performed a bivariate comparison of all factors? The p-values seem to refer only to one comparison. Please describe and adjust the presentation of analyses.

R: In Table 3, it is specified in the text (lines 202 to 205), in the table and at the bottom of Table 3 (line 215 and 216), that the statistical difference in yeast colonization on hands is analyzed between the 3 different study cycles (Clinical vs. Basic and Clinical vs. Preclinical).

In Table 6, we explain that the significant difference occurs in the use of moisturizing cream 2 or more times per day compared to 1 time per day or less (lines 243 to 246). It is also indicated in the table and at the bottom of the table (lines 253 to 254).

12.  Discussion: Authors mention biosafety and infection control practices among students. Please describe these measures.

R: The authors corrected this paragraph, removing the word "biosafety" and describing the infection control measures studied (lines 417 to 420).

13.  Lines 312-318: could more frequent application of moisturizing cream also be indirectly associated with more frequent hand hygiene? Please comment.

R: We agree with the reviewer and added this suggestion to the explanation for the use of moisturizer (lines 364 to 371)

14.  Please add a paragraph acknowledging the limitations of the study.

R: we add a paragraph indicating the recommendations for furthers studies according to our limitations (Lines 414 to 417).

Round 2

Reviewer 1 Report

Comments and Suggestions for Authors

Dear authors,

Thank you for considering my comments and making significant corrections to the manuscript. The introduced modifications significantly increase the vertical value of your publication, make it easier to read and understand, and provide new and valuable information.

The article in its current form, I recommend to be accepted for publication.

Author Response

Many thanks to reviewer 1 for his support and help

Reviewer 2 Report

Comments and Suggestions for Authors

Thank you for providing a revised version of your manuscript and for addressing my comments.

Concerning the limitations paragraph; this is more of a "future implications" description, based on their findings. However, I'd like to request that authors acknowledge some limitations related to their own study, eg.the methodology etc.

Comments on the Quality of English Language

Still needs improvements.

Author Response

  1. Concerning the limitations paragraph; this is more of a "future implications" description, based on their findings. However, I'd like to request that authors acknowledge some limitations related to their own study, eg.the methodology etc.

R: we incorporate the limitations (lines 413 – 415). “Among the limitations of our study are the absence of a control group of students, the lack of quantitative culture analysis, and the exclusion of other antifungal agents, such as fluconazole and amphotericin B”.
